# Characterization of RAP Signal Patterns, Temporal Relationships, and Artifact Profiles Derived from Intracranial Pressure Sensors in Acute Traumatic Neural Injury

**DOI:** 10.3390/s25020586

**Published:** 2025-01-20

**Authors:** Abrar Islam, Amanjyot Singh Sainbhi, Kevin Y. Stein, Nuray Vakitbilir, Alwyn Gomez, Noah Silvaggio, Tobias Bergmann, Mansoor Hayat, Logan Froese, Frederick A. Zeiler

**Affiliations:** 1Department of Biomedical Engineering, Price Faculty of Engineering, University of Manitoba, Winnipeg, MB R3T 2N2, Canada; amanjyot.s.sainbhi@gmail.com (A.S.S.); steink34@myumanitoba.ca (K.Y.S.); vakitbir@myumanitoba.ca (N.V.); bergmant@myumanitoba.ca (T.B.); frederick.zeiler@umanitoba.ca (F.A.Z.); 2Undergraduate Medicine, Rady Faculty of Health Sciences, University of Manitoba, Winnipeg, MB R3T 2N2, Canada; 3Section of Neurosurgery, Department of Surgery, Rady Faculty of Health Sciences, University of Manitoba, Winnipeg, MB R3T 2N2, Canada; gomeza35@myumanitoba.ca (A.G.); mansoor.hayat@umanitoba.ca (M.H.); 4Department of Human Anatomy and Cell Science, Rady Faculty of Health Sciences, University of Manitoba, Winnipeg, MB R3T 2N2, Canada; silvaggn@myumanitoba.ca; 5Department of Clinical Neurosciences, Karolinksa Institutet, 171 77 Stockholm, Sweden; log.froese@gmail.com; 6Pan Am Clinic Foundation, Winnipeg, MB R3M 3E4, Canada; 7Division of Anaesthesia, Department of Medicine, Addenbrooke’s Hospital, University of Cambridge, Cambridge CB2 1TN, UK

**Keywords:** traumatic brain injury, cerebral compliance, RAP, Signal Processing, Cerebral Dynamics

## Abstract

Goal: Current methodologies for assessing cerebral compliance using pressure sensor technologies are prone to errors and issues with inter- and intra-observer consistency. RAP, a metric for measuring intracranial compensatory reserve (and therefore compliance), holds promise. It is derived using the moving correlation between intracranial pressure (ICP) and the pulse amplitude of ICP (AMP). RAP remains largely unexplored in cases of moderate to severe acute traumatic neural injury (also known as traumatic brain injury (TBI)). The goal of this work is to explore the general description of (a) RAP signal patterns and behaviors derived from ICP pressure transducers, (b) temporal statistical relationships, and (c) the characterization of the artifact profile. Methods: Different summary and statistical measurements were used to describe RAP’s pattern and behaviors, along with performing sub-group analyses. The autoregressive integrated moving average (ARIMA) model was employed to outline the time-series structure of RAP across different temporal resolutions using the autoregressive (*p*-order) and moving average orders (*q*-order). After leveraging the time-series structure of RAP, similar methods were applied to ICP and AMP for comparison with RAP. Finally, key features were identified to distinguish artifacts in RAP. This might involve leveraging ICP/AMP signals and statistical structures. Results: The mean and time spent within the RAP threshold ranges ([0.4, 1], (0, 0.4), and [−1, 0]) indicate that RAP exhibited high positive values, suggesting an impaired compensatory reserve in TBI patients. The median optimal ARIMA model for each resolution and each signal was determined. Autocorrelative function (ACF) and partial ACF (PACF) plots of residuals verified the adequacy of these median optimal ARIMA models. The median of residuals indicates that ARIMA performed better with the higher-resolution data. To identify artifacts, (a) ICP *q*-order, AMP *p*-order, and RAP *p*-order and *q*-order, (b) residuals of ICP, AMP, and RAP, and (c) cross-correlation between residuals of RAP and AMP proved to be useful at the minute-by-minute resolution, whereas, for the 10-min-by-10-min data resolution, only the *q*-order of the optimal ARIMA model of ICP and AMP served as a distinguishing factor. Conclusions: RAP signals derived from ICP pressure sensor technology displayed reproducible behaviors across this population of TBI patients. ARIMA modeling at the higher resolution provided comparatively strong accuracy, and key features were identified leveraging these models that could identify RAP artifacts. Further research is needed to enhance artifact management and broaden applicability across varied datasets.

## 1. Introduction

Acute biomechanical traumatic neural injury, also termed traumatic brain injury (TBI), is a significant global health concern, causing over 50 million cases annually and incurring worldwide costs of approximately CAD 540 billion [1]. In Canada and globally, TBI remains a leading cause of death and disability [2]. The impact of moderate to severe TBI involves both primary and secondary injuries. Primary injuries occur at the moment of impact, causing immediate structural brain damage. In contrast, secondary injuries develop over time through systemic and cellular processes that exacerbate brain tissue damage. Unlike primary injuries, secondary injury mechanisms may respond to therapeutic interventions, offering opportunities to enhance patient outcomes. To prevent secondary injury across patient populations, current management strategies for moderate to severe TBI focus on guideline-based interventions that target physiological parameters using data from invasive pressure sensor technologies [2,3,4,5]. A key focus is maintaining intracranial pressure (ICP) below 22 mmHg, triggering therapeutic measures when exceeded [2,3]. ICP, which is often derived from invasive strain-gauge pressure sensors, is also used as an indicator of intracranial compliance, with the bedside manual visual inspection of pulse waveform morphology for assessing compensatory reserve. Intracranial compliance/compensatory reserve is a parameter that provides insight into the brain’s ability to adapt to changes in volume while maintaining stable pressure levels [6,7]. However, these methods are prone to errors, along with inter-observer and intra-observer consistency issues.

As a result, the RAP index was derived using signal sources from ICP pressure sensors, and has the potential for usage in TBI. RAP is a metric of intracranial compensatory reserve (and therefore compliance) derived using the moving correlation between ICP and the pulse amplitude of ICP (AMP) from any ICP pressure sensor technology [8,9,10,11,12]. In recent hydrocephalus studies, RAP (the correlation [R] between AMP [A] and ICP [P]) helped predict shunt failure in patients [8,9,10,11]. In addition, this index can be continuously calculated at the bedside in those patients with continuous ICP monitoring, optimally positioning it for use in TBI monitoring. As RAP values are the Pearson correlation coefficients, they range from −1 to +1, with lower positive values indicating good compliance, while higher positive and negative values suggest poor and exhausted compliance, respectively [8,9]. However, RAP has not been thoroughly investigated in moderate to severe TBI populations. Specifically, there is a lack of understanding of the general statistical behaviors of RAP in relation to ICP and AMP, its temporal time-series structure, and the characterization of its artifact profiles [7,13].

Therefore, this study aims to explore the following: (A) the general description of RAP signal patterns and behaviors, (B) the temporal statistical profile of RAP, and (C) the characterization of RAP artifact profiles. Gaining more profound insights into these aspects is essential for advancing the future integration of the RAP index into bedside monitoring, enhancing patient trajectory modeling, and supporting clinical intervention studies based on RAP values.

## 2. Materials and Methods

### 2.1. Patients

As with previous studies from our lab group [14,15], the data were retrospectively obtained from the TBI database prospectively maintained at the Multi-omic Analytics and Integrative Neuroinformatics in the HUman Brain (MAIN-HUB) Lab at the University of Manitoba. This study included patient data collected from January 2018 to March 2023. All patients in this cohort experienced moderate to severe TBI (Glasgow Coma Score < 12). Invasive ICP and arterial blood pressure (ABP) monitoring were conducted as per Brain Trauma Foundation (BTF) guidelines [2].

### 2.2. Ethics

Data collection was conducted with full approval from the University of Manitoba Health Research Ethics Board (H2017:181, H2017:188, and H2024:266).

### 2.3. Data Collection

In line with our previous work [14,15], all physiological data were recorded and digitized at a high frequency of 100 Hz or higher using Intensive Care Monitoring ‘Plus’ (ICM+ v8.5.4.6) data acquisition software, with analog-to-digital converters (Data Translations, DT9804 or DT9826) employed as needed. ABP was captured via radial arterial lines, while ICP was measured invasively using intra-parenchymal strain gauge probes (Codman ICP MicroSensor; Codman & Shurtleff Inc., Raynham, MA, USA) placed in the frontal lobe or using external ventricular drains (Medtronic, Minneapolis, MN, USA) in four cases.

For this study, demographic information at admission was extracted according to existing prognostic models in TBI. The collected demographic data included age, biological sex, admission pupillary response (bilaterally reactive, unilaterally reactive, or bilaterally unreactive), Marshall computed tomography (CT) grade, and Glasgow Outcome Scale-Extended (GOSE) grade.

### 2.4. Signal Processing

Post-acquisition processing of the above signals was conducted using ICM+, in keeping with our previously published methodology. ICP and ABP were initially decimated using 10 s moving averages updated every 10 s to avoid data overlap [14,15,16]. Mean arterial pressure (MAP) was subsequently calculated from ABP. AMP was obtained through Fourier analysis of the fundamental harmonic of the ICP waveform [7,17,18]. RAP was derived via the moving Pearson correlation coefficient between 30 consecutive 10 s mean windows (i.e., each calculation window was 5 min) of the parent signals (ICP and AMP), updated every minute according to previously validated methods [9,19,20,21]. This analysis also included cerebrovascular reactivity. The pressure reactivity index (PRx) is a continuous measure for assessing cerebrovascular reactivity [6,22,23]. Likewise, PRx was determined using the Pearson correlation coefficient between ICP and MAP, where the update period (i.e., one minute) and the calculation window size (i.e., 5 min) were similar to those of RAP [14,24,25].

### 2.5. Analysis of the Patterns and Behaviours of RAP

Alongside RAP, the analysis also included ICP, MAP, AMP, and CPP signals, since ICP and AMP were used to derive RAP [8,19], while MAP and cerebral perfusion pressure (CPP) helped establish standard thresholds used in RAP analysis in this field [2,26]. Firstly, Panda’s (a Python library) [27] describe function [28] from Python was used to find the summary measurements for each signal in all patients. Following this, a custom script was executed to find the time spent on RAP within certain threshold ranges (0.4 to 1, 0 to 0.4, and −1 to 0), based on a systematic review study previously conducted by our lab [19]. Afterwards, a comparative sub-group analysis was conducted based on age, biological sex, pupillary response, Marshall CT grade, outcome (GOSE grade), ICP, AMP, and PRx values. Threshold lines for these comparisons were established using commonly referenced values from prior studies [29,30,31,32] in related fields, as follows:Age—less than 40 years, 40 to 60 years, and above 60 years;Pupillary response—bilateral reactive, bilateral unreactive, and unilateral unreactive;Marshall CT grade—grade II, grade III, grade IV, and grade V;Outcome GOSE grade—alive/dead (2 or higher vs. 1) and favorable/unfavorable (5 or higher vs. 4 or less);ICP thresholds—below 20 mmHg and above 22 mmHg;AMP thresholds—below 1, between 1 and 3, and above 3;PRx thresholds—less than 0 vs. greater than 0 and less than 0.25 vs. greater than 0.25

Mann–Whitney U-test was utilized for the formal comparison since none of the groups showed normal distributions. One-way ANOVA was used to compare more than two groups. To run these operations, Python’s (version 3.7.16) mannwhitneyu [33] and f_oneway [34] functions from scipy.stats library were used, respectively.

### 2.6. Analysis of RAP Time-Series Structures

#### 2.6.1. Application of ARIMA Model

The autoregressive integrated moving average (ARIMA) model is a widely used statistical method for time-series forecasting [35,36,37,38]. It works by combining three main components, as follows: autoregression (AR), differencing to make data stationary (I for Integrated), and a moving average (MA). The model aims to capture the underlying patterns in time-series data and predict future values based on historical observations [35,36,37,38]. The AR part is controlled by the parameter *p*, representing the number of lagged observations in the model. It refers to the regression of the variable on its own lagged (previous) values. The parameter *d* represents the number of times the data need to be differenced to achieve stationarity. The MA part is controlled by parameter *q*, representing the number of lagged error terms in the model. It refers to modeling the error (or residual) term as a linear combination of previous error terms [35,36,37,38]. An ARIMA model is usually written as ARIMA(*p*, *d*, *q*). Assuming the signal is stationary (*d*-order = 0), a general autoregressive moving average model for a physiological signal, *X*, can be represented using Equation (1). In this model, *p* is the autoregressive order, *q* is the moving average order, *X_t_* is the signal at time *t*, *X_t_*_−*i*_ is the signal at time *t* − *i*, *ε_t_* is the error at time *t*, *ε_t_*_−*j*_ is the error at time *t* − *j*, *φ* is the autoregressive coefficient at time *t* − *i*, and *θ* is the moving average coefficient at time *t* − *j* [14].(1)$$X_{t}=c+\varepsilon_{t}+\sum_{i=1}^{p}\varphi_{i}X_{t-i}+\sum_{j=1}^{q}\theta_{j}\varepsilon_{t-j}$$

This ARIMA model was employed to capture the structure of time-series signals. ARIMA was chosen since it provides interpretability in terms of temporal dependencies (*p*, *d*, *q*), making it particularly suited for understanding signal dynamics, comparing temporal structure among signals and identifying artifacts. It is also a similar methodology to those used for determining the more basic aspects of cerebral blood flow physiologies [14,15]. According to previous works from the lab, both the *p*-order and *q*-order for determining the optimal ARIMA model varied from 0 to 10 [14,15]. The analysis was run on the differenced data (discussed in Section 2.6.3), and therefore, the *d*-order was set to 0, which effectively acted as *d*-order = 1. The ARIMA function from the statsmodels module [39] of Python was used for this analysis. Each combination of the orders was evaluated to find the optimal model for each signal and each patient.

#### 2.6.2. Statistical Metrics for ARIMA Analysis

Akaike Information Criterion (AIC), Bayesian Information Criterion (BIC), and Log-Likelihood (LL) were calculated to assess whether the models effectively captured the structure of the signal. These are statistical metrics used to evaluate the quality and goodness-of-fit of an ARIMA model, helping to assess how well the model captures the underlying structure of the time-series data [14,40,41]. Each of these metrics has its own characteristics and significance in model selection. AIC measures the goodness-of-fit of a model while penalizing for model complexity (the number of parameters). It balances model fit and complexity to avoid overfitting. BIC is similar to AIC but applies a more substantial penalty for models with more parameters, making it more conservative in terms of model complexity. LL measures the likelihood that the model could have generated the observed data. It reflects the fit of the model without penalizing for complexity. Lower AIC and BIC values indicate a better model, while higher LL values indicate a better fit [14,40,41].

According to a previous study from our lab, BIC is more stringent than AIC and LL [14]. On the other hand, models based on LL were more complex and could potentially overfit the data, leading to better residuals [14]. Given this, AIC was considered the most balanced option for model selection, as it strikes a middle ground between the stringency of BIC and the leniency of LL. For this reason, AIC was chosen to find the optimal ARIMA models for the signals.

#### 2.6.3. Stationarity Analysis

Since the ARIMA model is built on the assumption of the stationarity of time series, the data need to be stationary to apply an ARIMA model. Hence, stationarity analysis was carried out. Like in the previous work from our lab [16], Augmented Dickey–Fuller (ADF) and Kwiatkowski–Phillips–Schmidt–Shin (KPSS) tests were used to check the stationarity of all the signals. The ADF test indicates whether a time series is trend-stationary, while the KPSS test determines if the series remains stationary around a linear trend [36,42]. If the *p*-value from the ADF test is less and the *p*-value from the KPSS test is higher than a certain threshold, the time series is considered stationary. In line with previous studies [16], the threshold was set at 0.05 for both tests. The adfuller and kpss functions from the statsmodels module [39] in Python were used to perform the tests at the patient level. Additionally, due to the ADF and KPSS test results on the original data (discussed in Section 3.2.1), these tests were performed on each patient’s first-order-differenced data. It is noteworthy that the differenced data were achieved after temporal resolution.

#### 2.6.4. Generation of Different Temporal Resolutions of Data

ICP, AMP, and RAP were calculated across various temporal resolutions for a comprehensive analysis and to examine the impact of the temporal resolution reduction on the results. Subsequently, the optimal ARIMA model at the patient level was calculated for each parameter for every temporal resolution. The temporal resolutions applied in this study included minute-by-minute, 10 min intervals, 30 min intervals, and hour-by-hour intervals. The primary derived data were at minute-by-minute intervals. Afterward, Panda’s resample function [43] was used to reduce the resolution. The primary data (i.e., minute-by-minute resolution) mean of 10 min data points was determined as a single non-overlapping point in 10 min intervals. Similarly, for the 30 min and 1 h intervals, the means of 30 consecutive data points and 60 consecutive data points, respectively, were used to represent each interval.

#### 2.6.5. Evaluation Tools

After deriving the optimal ARIMA model for each patient, the median optimal ARIMA model for each signal was calculated based on those models (i.e., median *p*-order and median *q*-order values were calculated). The choice of the median value for optimal models across the whole population ensured a more representative summary across the examples while reducing the impact of outliers. To confirm the adequacy of the median optimal ARIMA models, the magnitude of residuals, autocorrelation function (ACF), and partial ACF (PACF) plots of residuals were examined by comparing the raw data to the modeled data [14,36]. Residuals represent the differences between actual data points and model-predicted values. The ACF measures the relationship between a time series and its past values, while the PACF indicates the correlation between a time series and its lagged values, excluding the effects of intermediate lags. For a well-fitted ARIMA model, residuals should be minimal, and the ACF and PACF plots should show no significant spikes at any lags, indicating that the model has captured the underlying structure [14,36]. Additionally, this analysis included the calculation of the overall variance in data, the residual variance, and the count of significant spikes to justify that the data had been modeled well.

### 2.7. RAP Artifact Segment Analysis

#### 2.7.1. Separating True Artifact Segments

For this section, true artifacts had to be calculated. Previously, to obtain clean and artifact-free data, artifacts were manually detected and removed from the raw collected data by experts in cerebral physiologic signal analysis and neurophysiology. Therefore, while comparing clean data with non-clean data, any additional data present in the non-clean version but absent in the clean version should be taken as representing artifacts. This step was performed by comparing timestamps of clean and non-clean data. Afterwards, identified artifact segments were saved into different comma-separated value (CSV) files.

#### 2.7.2. Analysis of Clean Data and Artifact Segments

The optimal models for the clean signal have already been obtained in the previous section. The optimal models for the artifact segments were calculated for each signal of each patient using the same methodology. Finally, a comparison between clean data and artifact segments based on temporal structure was conducted using various statistical techniques. The temporal resolutions applied in this analysis included minute-by-minute and 10 min intervals. The remaining resolutions were excluded from this analysis because, at such low resolutions, the quantity of artifact data would be insufficient to yield significant results in this section.

This analysis focused on three key areas:(i)Comparing optimal ARIMA models—The optimal models for the clean data were obtained in the previous section (i.e., Section 2.6). The optimal models for the artifact segments were also computed for each signal of each patient using the same methodology. These models for clean data and artifact segments are expected to differ, resulting in varying *p*-orders and *q*-orders between the two groups for each patient. A formal comparison of the groups’ ARIMA orders for each signal was conducted using the Mann–Whitney U test, with scatterplots provided for visual representation;(ii)Comparing the residuals—Using the median optimal ARIMA model calculated for clean data in Section 2.6, residuals for clean and artifact segments were computed and formally compared at the patient level. The results are expected to indicate significant differences in the mean residuals and variance of residuals between the clean and artifact groups;(iii)Comparing the cross-correlation of residuals—If cross-correlation is calculated between RAP residuals and ICP/AMP residuals, the expectation is that the maximum correlation value between clean RAP and clean ICP/AMP residuals would be higher than that between clean RAP and artifact ICP/AMP residuals. Since RAP is derived from ICP and AMP, their residuals should naturally show a strong correlation. However, this correlation is expected to decrease when considering the artifact segments of ICP/AMP, as these segments do not accurately represent true ICP/AMP values. The *correlate* function from the *Numpy* library was used to calculate the cross-correlations, and it measured the similarity between two signals (or datasets) as a function of the time lag applied to one of them (i.e., calculated dot product).

#### 2.7.3. Evaluating Identified Features

After analyzing the data to identify features with the potential to effectively distinguish artifacts, a simple sliding window approach was applied to the non-clean data to assess the success rate of these features in identifying artifacts within the signal. The success rate of capturing artifacts within the signal was calculated as (captured artifacts/true artifacts) × 100%.

## 3. Results

### 3.1. Patient Demographics

As reported in Table 1, 109 TBI patients were included in this study, with a median recording duration of 4125.13 min. The median age of the patients was 43 years (interquartile range (IQR): 29 to 57), and 89 of the patients were male (81.65%). The median Glasgow Coma Scale (GCS) score was 7 (IQR: 4 to 8), while the median motor sub-score was 4 (IQR: 2 to 5).

### 3.2. General RAP Patterns and Behaviours

#### 3.2.1. Summary Measurements

There were some unrealistic values of ICP and MAP in the data that could lead to erroneous CPP, AMP, and RAP values, since they are derived from them. Therefore, according to the previous studies [44], data points with ICP > 100 mmHg or <−15 mmHg and MAP > 200 mmHg or <0 mmHg were excluded from this analysis. Afterwards, summary measures of the aforementioned parameters were calculated and are depicted in Table A1 of Appendix A. Notably, RAP had a mean of 0.632 ± 0.483.

Based on our previous study, RAP was classified into three distinct states according to its value [19], which were as follows: (i) state 1, representing a healthy condition, was characterized by small positive RAP values close to zero; (ii) state 2, which was most commonly observed in TBI patients, reflected impaired intracranial compliance and compensatory reserve, with elevated RAP values (RAP > 0.4), and (iii) state 3 occurred in more severe conditions, indicating the further deterioration of compensatory reserve, cerebrovascular reactivity, and cerebral autoregulation. This state was associated with a significant number of fatal outcomes and was marked by declining RAP values, including, in some cases, negative RAP [19]. Based on the mean RAP obtained in this study, it can be stated that the mean RAP fell into state 2, indicating impaired cerebral compliance and compensatory reserve. This finding is consistent with our previous study, wherein state 2 was also the most commonly observed condition among TBI patients [19].

#### 3.2.2. Time Spent Within Thresholds

According to the threshold ranges outlined in Section 2.5, the percentage of time spent within each range was calculated for all patients and is presented in Table A2 of Appendix A. So, using the RAP index, the highest percentage of time spent was in the impaired state (ranging from RAP of 0.4 to 1), which was 78.091% and corresponds to state 2, as previously defined [19].

#### 3.2.3. Sub-Group Analysis

The sub-group analysis results are shown in Appendix A Table A3, Table A4, Table A5, Table A6, Table A7, Table A8, Table A9, Table A10, Table A11, Table A12, Table A13, Table A14 and Table A15. In the age groups analysis, for the first age comparison (i.e., age above and below 40 years), only ICP showed a significant difference (*p* = 0.048) between the two groups, with AMP having a near-significant *p*-value of 0.065227, as depicted in Table A3 of Appendix A. In the second age comparison (age below 40 years, 40 to 60 years, and above 60 years), AMP and RAP were significantly different (*p* = 0.046, *p* = 0.005), while ICP showed no significant difference (*p* = 0.284) (Table A4 of Appendix A).

In the M/F sex groups, only CPP (*p* = 0.008) was significantly different (Table A5 of Appendix A). For pupillary response, none of the parameters showed significant differences (Table A6 of Appendix A). In Marshall CT grade groupings, ICP, AMP, and RAP were significantly different across groups (*p* = 0.002, *p* = 0.0003, *p* = 0.00001) (Table A7 of Appendix A). RAP increased from grade II to IV, but at grade V, it decreased. A similar case was observed for ICP and AMP.

For outcome comparison, the GOSE grade was assessed at 1-month and 6-month intervals (Table A8, Table A9, Table A10 and Table A11 of Appendix A). In the alive/dead comparison, for both cases, AMP significantly differentiated the groups in both intervals (*p* = 0.003, *p* = 0.01), while ICP and CPP were only significant in the 1-month GOSE results (*p* = 0.033, *p* = 0.032). However, RAP was not significant in either case. In the favorable/unfavorable cases, ICP, CPP, and AMP were significantly different for both intervals, but RAP again showed no significant difference in either case (*p* = 0.294, *p* = 0.403).

In the subgroup analysis for ICP, all parameters showed significant differences between the two groups (*p* = close to 0 for both groups). The group with ICP > 22 mmHg had higher RAP and AMP values than the other group. Similarly, in the AMP analysis, both ICP and RAP increased as AMP rose, reflecting the findings from the ICP subgroup analysis, with all parameters displaying significant differences.

Finally, in PRx analysis, the first comparison showed that RAP had a lower mean value when PRx > 0 compared to PRx < 0, though ICP and AMP were higher. This finding suggests that impaired cerebrovascular reactivity (PRx > 0) was associated with reduced RAP, aligning with the results of our systematic review [19]. This decrease in RAP corresponded to state 3 of RAP [19], as defined in Section 3.2.1. Similarly, in the second comparison, using a PRx threshold of 0.25, a comparable pattern emerged, with lower RAP being associated with higher PRx values (i.e., PRx > 0.25). Similar to the ICP and AMP threshold analyses, all parameters demonstrated significant differences between sub-groups in the PRx threshold analysis (*p* = close to 0 for all cases). The results of these three threshold analyses are presented in Table A12, Table A13, Table A14 and Table A15 of Appendix A.

### 3.3. Optimal ARIMA Structure Analysis

#### 3.3.1. Stationarity Assessment

As discussed in Section 2.6.3, ADF and KPSS tests were performed for each patient and signal to check the stationarity of the signals. Initially, these tests were applied to the original data. Table A16 and Table A17 of Appendix B illustrate the *p*-values for each patient’s test results at the minute-by-minute data resolution. Additionally, Table A20 and Table A21 of Appendix C show the summarized results. As shown in the tables, while most of the signals appeared stationary according to the ADF tests, the KPSS test indicated that most were non-stationary. This suggests that the signals are largely trend-stationary, but likely non-stationary around a linear trend. However, for ARIMA model analysis, the data need to be stationary in terms of both cases. Therefore, a first-order difference was applied to the original data after temporal resolution. Table A18 and Table A19 of Appendix B show the *p*-values for each patient’s test results at the minute-by-minute data resolution. The resulting outcomes are summarized in Table A22 and Table A23 of Appendix C.

As evident from the tables, after applying first-order differencing, nearly all the data, with a few exceptions, were assessed as stationary in both tests. Therefore, these differenced data were suitable for ARIMA model analysis.

#### 3.3.2. Determination of Optimal ARIMA Models

Appendix D provides the optimal ARIMA models for each signal at each resolution for each patient, detailing the *p*-, *d*-, and *q*-orders along with their AIC values. Based on these results, the population global median optimal models in each resolution were calculated and are shown in Table 2. As shown in the table, the models for each signal were quite similar in the case of 10-min resolution and below.

#### 3.3.3. Evaluation of Optimal ARIMA Models

With the median optimal ARIMA models now determined, the quality of these models can be assessed using the residuals, ACF, and PACF plots of residuals. Figure 1 represents a patient example of the ACF and PACF plots of residuals for RAP at 1 min intervals for pre- and post-ARIMA modeled data.

The figure corresponds to the residuals of the RAP signal (a) before and (b) after ARIMA. The original plots had significant spikes in both ACF and PACF plots, while the spikes in post-ARIMA (3, 1, 3) were mostly within the 95% confidence interval, indicating that the model moderately accounts for the RAP structure.

ACF, autocorrelative function; ARIMA, autoregressive integrated moving average; PACF, partial autocorrelative function; RAP, compensatory reserve index.

Figure 1a corresponds to the original RAP data since the orders were set to zero (ARIMA (0, 0, 0)). Figure 1b utilizes the median optimal ARIMA model for RAP (ARIMA (3, 1, 3)). It is evident that the ACF plot of the residuals using the original data showed a gradual decay, while the PACF plot had significant spikes at various lags, indicating that the ARIMA (0, 0, 0) model did not effectively capture the signal’s structure. On the contrary, after applying the median optimal ARIMA model for RAP (3, 1, 3), only two significant spikes were present at lag 15, with another after lag 30 in the PACF plot. Otherwise, most of the ACF and PACF values fell within the 95% confidence interval, implying that any autocorrelation left in the residuals is not statistically significant and suggesting that the calculated ARIMA model successfully captured the structure of the signal.

Figure A1 and Figure A3 of Appendix E illustrate a patient example of ACF and PACF plots with this comparison for ICP and AMP signals, and demonstrate similar results, proving that the calculated median optimal ARIMA model captures the time-series structure with moderate performance.

While visually, the performance of the optimal ARIMA model is satisfactory, the global population median of residuals was compared with it to provide a better description between the original data and those yielded after optimal ARIMA model application for all signals, as reported in Table 3.

While calculating, the absolute value of the residuals of each data point was taken for the betterment of the calculation. The median residual of the optimal ARIMA model was substantially less than that of the original data for all signals, which further emphasizes the ARIMA model’s success in capturing the signal structure. Additionally, the variance of the overall data, variance in the residuals, and the number of significant spikes were calculated and are shown in Table A25 of Appendix E. It can be seen that the variance of the residuals was smaller than that of the original data. Furthermore, both ACF and PACF plots for the modeled data had only one significant spike, in contrast to the original ACF and PACF plots, which had seven and two spikes, respectively. These values align with the results from Figure 1 and Table 3, further proving that the data are being modeled. The values of these parameters were calculated for each signal of each patient at the minute-by-minute resolution, as shown in Table A27, Table A28 and Table A29 of Appendix E. Subsequently, the means and medians were determined, as presented in Table A30 and Table A31 of Appendix E. The attribute values in these tables indicate that the data were adequately modeled. The values of the variance in the residuals and the number of significant spikes in ACF and PACF plots were much smaller than those of the original data.

A similar analysis was performed at different temporal resolutions for the same patients. Figure 2 shows a patient example of the ACF and PACF plots of the residuals for RAP at the remaining temporal resolutions. Even though all the lags were within the 95% confidence interval at all the resolutions (which was also justified by the result in Table A26 of Appendix E), the ACF and PACF plots exhibited comparatively significant spikes with higher magnitudes than those at the minute-by-minute temporal resolution (depicted in Figure 1b). This indicates that while the optimal ARIMA model captures the time-series structure in both cases, it is more successful at the minute-by-minute resolution than at other lower temporal resolutions. Figure A2 and Figure A4 of Appendix E present these comparative figures for ICP and AMP at the 10 min, 30 min and 1 h intervals, which also displayed similar characteristics in the ACF and PACF plots of the residuals.

The figure documents the ACF and PACF of the residuals of the RAP-mapped ARIMA structure in the (a) 10-min-by-10-min, (b) 30-min-by-30-min, and (c) hour-by-hour relationships.

ACF, autocorrelative function; ARIMA, autoregressive integrated moving average; PACF, partial autocorrelative function; RAP, compensatory reserve index.

### 3.4. Assessment of the Features for Identifying Artifacts

#### 3.4.1. Comparing Optimal ARIMA Models

Initially, the optimal ARIMA models for the artifact segments of each signal of each patient were calculated. The results are depicted in Table A33 of Appendix F. Afterwards, the medians and means of the orders of the ARIMA models for the two groups were calculated. Table A34 and Table A35 of Appendix F contain the median and the mean results for the 1 min and 10 min temporal resolutions. *d*-order was not included in the analysis, as it was set to 1 across all cases. While calculating optimal models for the artifact segments of 10 min data, five examples (i.e., patients) could not provide any result because of the inadequacy of data (i.e., artifact segments). As both tables show, the median and mean orders of all clean and artifact data parameters differed significantly, particularly for the minute-by-minute data. The Mann–Whitney U test that was conducted on these two groups of orders could provide a clear statistical comparison. The resulting *p*-values are as follows.

As reported in Table 4, since *p*-value < 0.05 indicates a significant difference between the two groups, it can be concluded that in the case of minute-by-minute data, both the *p*-orders (*p* = close to 0) and *q*-orders (*p* = 0.01526) of the RAP ARIMA model can serve as effective indicators for distinguishing artifact segments from clean data. However, only the *q*-order of ICP (*p* = 0.00058) and the *p*-order of AMP (*p* = 0.00501) demonstrated a significant difference between the two groups. For the lower-resolution data, RAP did not appear to show any notable differences across any orders. However, the *q*-orders of ICP (*p* = close to 0) and AMP (*p* = 0.00032) exhibited significant differences.

To visually represent the data, scatterplots comparing the orders of the two groups are shown in Figure 3. As seen in the figure, there are only a few patient examples wherein the clean and artifact segment orders overlap in the minute-by-minute resolution (Figure 3a). The majority of them differ, confirming that the optimal models for the parameters of clean and artifact segments are quite distinct. In the case of the 10 min resolution data, although many instances show non-overlapping orders, there are more examples of overlap compared to the minute-by-minute resolution. Scatterplots for RAP *q*-order and the rest of the signals are demonstrated in Appendix F Figure A6, Figure A7 and Figure A8. They also displayed similar results, with both of the orders differing between the two groups and the minute-by-minute resolution showing a more pronounced distinction.

The figure demonstrates the values of the *p*-orders from the ARIMA model of the clean vs. artifact for each patient at (a) minute-by-minute resolution and (b) 10-min-by-10-min resolution. The blue circles correspond to the *p*-orders of the cleaned data, whereas the red crosses represent the *p*-orders of the artifact segment. If a red cross overlaps a blue circle, the value of the order for that patient is the same. If they do not overlap, the values differ.

ARIMA, autoregressive integrated moving average; RAP, compensatory reserve index.

As seen from the figures, there are only a few patient examples wherein the clean and artifact segment orders overlap in the minute-by-minute resolution (Figure 3a). The majority of them differ, confirming that the optimal models for the parameters of clean and artifact segments are quite distinct. In the case of the 10-min resolution data, although many instances show non-overlapping orders, there are more examples of overlap compared to the minute-by-minute resolution. Scatterplots for RAP *q*-order and the rest of the signals are demonstrated in Appendix F Figure A6, Figure A7 and Figure A8. They also display similar results, with most of the orders differing between the two groups and the minute-by-minute resolution showing a more pronounced distinction.

To assess the success rate of artifact identification, the sliding window method was applied at the patient level. For minute-by-minute data, a window size of 100 with a sample size of 50 was used, while for 10-min-by-10-min data, a window size of 50 with a sample size of 25 was employed. Within each window, the optimal ARIMA model was calculated, and artifacts were identified in the window if any of the calculated orders (*p*, *d*, or *q*) differed by more than three from those of the clean data optimal model. Using this approach, the average success rates for artifact identification were 65.258% for ICP, 65.258% for AMP, and 84.038% for RAP at the minute-by-minute resolution. The average success rates for artifact identification at the 10-min-by-10-min resolution were 55.336% for ICP, 54.128% for AMP, and 43.089% for RAP. Table A36 and Table A37 of Appendix G depict the average success rates for artifact identification in both temporal resolutions.

#### 3.4.2. Comparing the Residuals

Using the median optimal model described in Table 5, the residuals of clean data and artifact segments were calculated for all patients. The difference between the two groups was determined using the Mann–Whitney U test for each patient. The result is summarized below, where significant corresponds to *p* < 0.05 and insignificant is otherwise.

As seen from the table, most cases showed significant differences at the minute-by-minute resolution while comparing clean residuals with artifact residuals. On the contrary, the result was the opposite in the 10-min temporal resolution, with the majority of examples belonging to the insignificant group.

While calculating residuals, a few patient examples had inadequate data points (1 patient at the minute-by-minute data resolution for ICP and 13 patients at the 10 min data resolution across all signals). Consequently, the residuals for those patients could not be calculated, and the analysis was carried out excluding them.

Additionally, to consider residuals as a feature to identify artifacts, the residuals of all the clean data should be low. In other words, they need to be consistent and should be fitted by the median optimal ARIMA model calculated for the clean data. To check this, the variance of the residuals of each patient was calculated (Table 6). The same was done for the artifact data (whose expected variance should be higher). The median and mean values of variance of each group were calculated as shown below.

As shown in the table, the expected outcome was observed for both resolutions. Particularly, ICP showed the largest difference among the signals. However, the median results of AMP at the 10-min-by-10-min data resolution deviated from the expected result.

The sliding window method was applied at the patient level to evaluate the success rate of artifact identification for this feature. A window size of 50 with a sample size of 25 was used for both minute-by-minute and 10-min-by-10-min resolution. At first, the variance of residuals within each window was calculated, and artifacts were identified if the variance of residuals exceeded the median variance of the total data for a single patient. Using this method, the average success rates for identifying artifacts across the entire population were 70.212% for ICP, 56.916% for AMP and 91.666% for RAP at the minute-by-minute resolution, and 85.092% for ICP, 74.264% for AMP and 84.411% for RAP at the 10-min-by-10-min resolution, as illustrated in Table A36 and Table A37 of Appendix G.

#### 3.4.3. Comparing the Cross-Correlation of Residuals

The groups for this analysis were formed as outlined in Section 2.7.2. After calculating the cross-correlation of total signals for each case and patient, the maximum values from the results were recorded. Next, the median and mean values of the maximum cross-correlation between RAP and ICP/AMP residuals were calculated across the total population for both the clean and artifact cases, as follows below.

Table 7 demonstrates that for the minute-by-minute data, the maximum RAP–AMP cross-correlation of residuals was expectedly higher in clean–clean cases compared to clean–artifact cases, based on both median and mean values. However, this was not the case for RAP–ICP; even though the mean value of clean–clean cases was slightly higher, the median was lower. On the other hand, for the 10-min-by-10-min data, none of the clean–clean cases had considerably higher values in RAP–ICP (median and mean) cross-correlation. In contrast, RAP–AMP cross-correlation showed higher values in clean–clean cases in terms of both median and mean.

A Mann–Whitney U test was subsequently performed to compare the two groups—maximum cross-correlation of clean RAP residuals with clean ICP/AMP residuals vs. clean RAP residuals with artifact ICP/AMP residuals. For RAP–ICP, the *p*-values were 0.02809 for the minute-by-minute resolution and 0.31919 for the 10 min resolution. In contrast, for RAP–AMP, the *p*-values were close to 0 for both resolutions.

Additionally, the maximum cross-correlations of the two groups (i.e., RAP–ICP clean–clean residuals vs. clean–artifact residuals and RAP–AMP clean–clean residuals vs. clean–artifact residuals) were compared at the patient level. The expected result was that the clean–clean maximum cross-correlation of residuals should be greater than the clean–artifact maximum cross-correlation of residuals, as explained in Section 2.7.2. The numbers of patients (out of 108 patients in total) in each case that showed greater values in clean–clean cases are as follows: (i) RAP–ICP, 34 cases and (ii) RAP–AMP, 95 cases at the minute-by-minute resolution; (i) RAP–ICP, 41 cases and (ii) RAP–AMP, 69 cases at the 10-min-by-10-min resolution.

The outcome of this analysis aligns with the findings of the overall (median and mean) result presented in Table 7, showing that the RAP–AMP cross-correlation at the minute-by-minute resolution demonstrated the greatest number of patients (95 cases) with higher values of maximum cross-correlation in clean–clean cases, alongside RAP–AMP cross-correlation at the 10-min-by-10-min resolution demonstrating 69 cases. On the contrary, RAP–ICP failed to achieve such large numbers at both resolutions.

In the success rate findings of this feature, RAP–ICP and RAP–AMP cross-correlations were calculated within each window. The window sizes and sample sizes were similar to the previous feature (i.e., 50 and 25, respectively). Artifacts were predicted within a window if the maximum cross-correlation was lower than the median of the maximum cross-correlation values between clean and artifact groups of the total recording for a single patient. Using this method, the median success rates for identifying artifacts across the entire population were 37.011% for RAP–ICP and 61.6% for RAP–AMP cross-correlation at the minute-by-minute resolution, whereas at the 10-min-by-10-min resolution, they were 6.512% and 35.829%, respectively.

## 4. Discussion

We set out to explore the RAP compensatory reserve index, derived from ICP pressure sensors, to better understand some critical aspects of such cerebral data streams. First, we comprehensively characterized the general nature of RAP signals with respect to other cerebral physiologic parameters, including subgroup analysis. Second, we outlined the time-series statistical structures of RAP in relation to its constituent signals (ICP and AMP). Finally, we leveraged our enhanced understanding of the time-series structures of RAP data streams to explore signal artifact detection. Throughout this process, some important aspects of RAP and the use of such sensor data streams deserve to be highlighted.

### 4.1. RAP’s Patterns and Behaviours

First, based on the results in Appendix A, TBI patients generally demonstrated an impaired compensatory reserve, as they spent most of their time within the range of 0.4 to 1, as measured by the RAP index (illustrated in Table A2 of Appendix A), corresponding to state 2 of RAP [19], as defined in Section 3.2.1. Regarding age comparison, Table A3 and Table A4 of Appendix A indicate that RAP increased with age. For Marshall CT grades, grades I through IV represent progressively worsening brain injuries [15]; thus, RAP would be expected to increase from grade I to IV, as supported by the analysis in Table A7 of Appendix A. In grade V, patients underwent brain surgery whereby mass lesions were removed [15]. This may or may not have resulted in a higher RAP than grade IV, depending on the surgery’s outcome, and can explain the reduced RAP value observed in grade V.

### 4.2. Time Series Structure Analysis

Second, during the time series analysis, it was clear that RAP signal sources were non-stationary and carried substantial trend features inherent within their data streams. We were able to demonstrate this across two different temporal resolutions of RAP data, emphasizing that this was present even at low temporal resolutions. This is critical for the future use of RAP in physiologic modeling, as not accounting for such a trend would lead to model inaccuracies, and most work in the field to date ignores such features.

RAP data streams displayed inherent autoregressive features, consistent with optimal ARIMA models with non-zero autocorrelative and moving average orders (*p*-orders and *q*-orders, respectively). Also of interest, the optimal ARIMA model orders for RAP differed from both ICP and AMP, and its constituent signals, highlighting that RAP contains different information compared to ICP or AMP alone. This was the case across the population, highlighting again the need to account not just for the data trend, but also for more complex autoregressive features, in future modeling using temporally resolved RAP data. However, it must be noted that the median optimal model calculated for the dataset may not accurately represent all patients. For instance, RAP at minute-by-minute resolution had a median optimal ARIMA model of (3, 1, 3). However, one patient individually obtained an optimal ARIMA model (8, 1, 3). Hence, applying the ARIMA (3, 1, 3) model to this patient’s RAP signal may not effectively capture the data’s structure due to the substantial difference in the *p*-order. Examining this patient’s ACF and PACF plots shown in Figure A5 of Appendix E reveals spikes between lags 0 to 5 that fall outside the confidence intervals, highlighting further limitations of using the median optimal ARIMA model. Nevertheless, the spikes out of the confidence intervals had very small magnitudes compared to the spikes seen in the ACF and PACF plots from the original data. Therefore, the median optimal ARIMA model obtained in this analysis could contribute to the identification of the features that helped distinguish clean data from artifact segments.

### 4.3. Comparison Among Different Resolutions

Third, during ARIMA model generation, including stationarity tests, some examples failed to return a *p*-value due to insufficient data points, which were most commonly observed at the hour-by-hour temporal resolution. Similarly, most of the non-stationary results were also found at this lowest resolution in both the original and differenced data. Additionally, while calculating the optimal ARIMA model for each signal of each patient, some cases failed to yield results due to insufficient data, mainly at lower resolutions. These observations highlight the critical role of data point quantity in each step of determining the optimal ARIMA model. It also suggests that lower resolutions may be associated with higher residuals, indicating a comparatively less accurate model. The medians of the residuals for each resolution were calculated with the results summarized in Table A32 of Appendix E. All of these findings emphasize the importance of a proper understanding of the statistical structures of such data streams from pressure sensors and their derived metrics (such as RAP). Throughout Table 2, the loss of RAP lags can be observed in the median optimal ARIMA models for lower resolutions (i.e., order numbers are smaller). Additionally, Table 5 and Table 6 show the lower importance of artifact management, since the difference between clean and artificial groups was not significantly different at lower resolutions. This suggests that the lower resolutions lose the dynamic aspects of the data (ICP/AMP/RAP).

This trend can also be observed in Table A24 of Appendix D, which details the optimal models for each signal and resolution for each patient. Lower resolutions tend to have simpler optimal models with lower order (*p*, *d*, *q*) values, leading to underfitting. This occurred because the ARIMA analysis, constrained by fewer data points, could not find a suitable model to capture the data fully. In contrast, higher resolutions, with more data points, yielded better results. The presence of spikes with higher magnitudes in the ACF and PACF plots of residuals at lower resolutions from Figure 2 and Appendix F further supports this statement.

### 4.4. Identifying Artifacts

Finally, building on the results from the time-series modeling of RAP, ICP, and AMP, we aimed to identify potential features capable of distinguishing artifacts from clean data. To qualify as a potential identifier of artifact profiles, a feature must show significant differences between the clean and artifact groups in every case; for instance, in a Mann–Whitney U-test analysis, the *p*-value between the two groups should be less than 0.05. Firstly, according to Table 4, the *p*-values for ICP *q*-order, AMP *p*-order, and RAP *p*-orders and *q*-orders were less than 0.05 (i.e., significant difference) while comparing optimal ARIMA models of clean and artifact data at the minute-by-minute resolution, suggesting that these orders are strong candidates for identifying artifacts. Conversely, ICP *p*-orders and AMP *q*-orders from the optimal models could be excluded as potential features due to insignificant differences between the clean and artifact groups (*p*-value > 0.05). However, at the 10-min-by-10-min data resolution, only the *q*-order of optimal ARIMA models of ICP (*p*-value = close to 0) and AMP (*p*-value = 0.00032) proved to be a potential distinguishing factor, while other orders showed no significant differences.

Secondly, while comparing the residuals of clean and artifact profiles at the patient level, Table 5 shows that the majority of cases exhibited significant differences for each signal at the minute-by-minute resolution. Specifically, significant differences were observed in 63, 64, and 65 cases out of 108 for ICP, AMP, and RAP, respectively. In contrast, at the 10-min-by-10-min resolution, most cases showed no significant differences for all signals. This indicates that though residuals could be a strong feature for distinguishing artifact profiles at the minute-by-minute resolution, they are less effective at the 10 min resolution. Table 6 further supports this finding, as the medians and means of the variance of residuals across the population were consistently lower for clean data and higher for artifact segments at both resolutions. This consistency suggests that clean data were moderately well-modeled. Therefore, residuals could be considered as a reliable feature for artifact identification.

The third and final analysis focused on comparing the maximum cross-correlation of residuals between clean–clean and clean–artifact combinations for RAP–ICP and RAP–AMP. This analysis was conducted in three parts. It was hypothesized that the maximum cross-correlation between clean RAP and clean ICP/AMP residuals would be higher than that between clean RAP and artifact ICP/AMP residuals, as RAP is derived from ICP and AMP, and their residuals are expected to exhibit strong correlations. Firstly, the medians and means of the maximum cross-correlations were calculated. Among these, only the RAP–AMP cross-correlation consistently showed higher values in clean–clean cases for both median and mean. Thus, the RAP–AMP maximum cross-correlation emerged as a potential feature for both minute-by-minute and 10-min-by-10-min resolutions. Secondly, a Mann–Whitney U test was performed across the entire population to compare the groups. The test revealed significant differences (*p* < 0.05) between clean–clean and clean–artifact residuals for each case except RAP–ICP at the 10-min-by-10-min resolution. Finally, maximum cross-correlations were compared at the patient level, with the expectation that clean–clean maximum cross-correlations would be greater than clean–artifact correlations. This was confirmed for RAP–AMP cross-correlation at both the minute-by-minute and 10-min-by-10-min resolutions, where the majority of patients exhibited higher clean–clean values. In conclusion, combining these three parts, between RAP–ICP and RAP–AMP, the cross-correlation of the latter at both resolutions could serve as a strong feature for identifying artifacts.

Fourthly, a detailed treatment-based sub-group assessment is required. This analysis did not include the effects of different therapeutic interventions, such as decompressive craniectomy, pCO_2_ changes or mannitol infusion. Though ICP treatments have an immediate impact on ICP (the minutes after treatment), their long-term impact on ICP modeling and other derived ICP measures (like PRx) is quite limited [45,46,47]. Therefore, when modeling and assessing RAP physiological factors over larger periods of time and over whole populations, ICP treatment factors can likely be largely ignored. However, when robust minute-by-minute RAP is being modeled (looking at individual moments of patient state), these factors should be considered.

Finally, while the success rate of capturing artifacts showed promising results, some non-artifact data points were mistakenly identified as artifacts (i.e., false positives). The number of false positives at each parameter and each analysis is demonstrated in Table A36 and Table A37 of Appendix G. Removing these non-artifact data points could result in the loss of valuable information from the signal. Further work is needed to address this issue, either by refining the thresholds and parameters in the sliding window approach, or by utilizing machine learning (ML) methods and incorporating these identified features into the model.

## 5. Limitations

The population sample size is relatively small despite representing the largest study to date comprehensively characterizing RAP data features. Such small sample sizes limit the ability to extrapolate such findings to other populations where ICP sensor technology is applied, and RAP can be measured. For instance, as discussed in the previous section, the median optimal model identified for the clean data may not fully represent all patients. Secondly, the number of data points for each patient constrains the results at lower resolutions. The reduction in data points at lower resolutions, due to calculation methods, led many optimal models at these resolutions to return ARIMA (1, 1, 1), indicating insufficient data points to capture the signal structure and resulting in underfitting. This is supported by the table showing higher residuals at lower resolutions (Table A31 of Appendix E). Additionally, several examples failed to return optimal ARIMA models, residuals, or *p*-values in the Mann–Whitney U-test, further decreasing the data size at lower resolutions. Moreover, it remains unclear why RAP at lower resolutions did not show significant order differences between clean and artifact groups, while RAP at higher resolutions did. This could be due to RAP’s derivation method, which results in 80% overlapping data, or due to insufficient data points at lower resolutions. Thirdly, the heterogeneity in TBI characteristics and the diversity of treatments administered could have influenced the physiological response observed in the signals, which might make it difficult to identify consistent patterns and draw generalized conclusions Finally, the data originate from a single-center archive, limiting generalizability, as findings may not apply to other centers with different patient populations, treatment protocols, or equipment.

## 6. Future Directions

Future work on ICP pressure sensor-based signal sources, including RAP, needs to include larger multi-center high-frequency signal databases. With improved sample sizes, the validation of the above general RAP behavior, its time-series structure and artifact detection methods need to occur. Such future work could include non-linear methods and future sub-group analysis based on injury or disease patterns. Further, artifact detection methods could be enhanced to include not just the time-series methods explored within this manuscript, but also layered approaches, including signal morphological assessments, wavelet decomposition methods, and ML techniques. Finally, for RAP data to be temporally modeled, a proper understanding of its time-domain statistical features is key. Such larger multi-center studies would be optimally positioned to define RAP statistical features more robustly.

## 7. Conclusions

RAP signals, derived from ICP sensor technology, displayed reproducible and characteristic patterns in this population of moderate/severe TBI patients, with most displaying features of impaired compensatory reserve. The time-series statistical features of RAP demonstrated inherent autoregressive features and data trends, regardless of temporal resolution. Such time-domain statistical features of RAP signals can be used to identify artifactual segments in RAP data streams. Future work is required in larger populations to validate such findings.

## Figures and Tables

**Figure 1 sensors-25-00586-f001:**
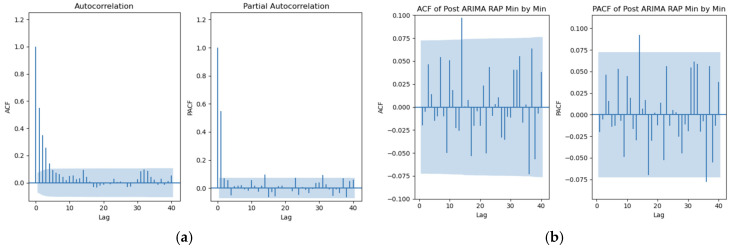
ACF and PACF plots at minute-by-minute temporal resolution—patient example. (**a**) RAP pre-ARIMA plots, (**b**) RAP post-ARIMA (3, 1, 3) plots.

**Figure 2 sensors-25-00586-f002:**
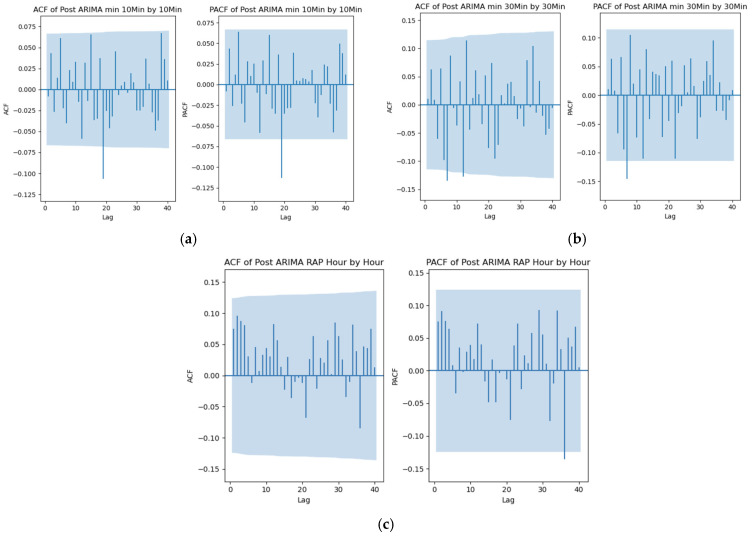
ACF and PACF plots at different resolutions—patient example. (**a**) At 10-min-by-10-min resolution with ARIMA (1, 1, 1), (**b**) at 30-min-by-30-min resolution with ARIMA (1, 1, 1), (**c**) at hour-by-hour resolution with ARIMA (1, 1, 1).

**Figure 3 sensors-25-00586-f003:**
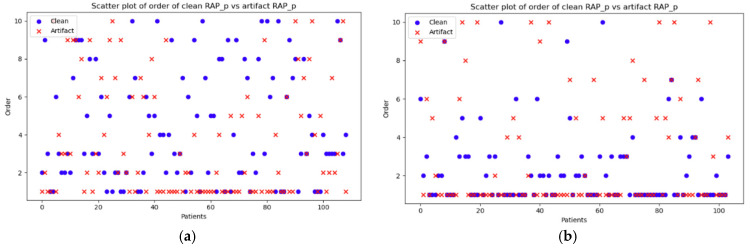
Scatterplots for RAP *p*-orders at different resolutions for each patient. (**a**) at minute-by-minute resolution, (**b**) at 10-min-by-10-min resolution.

**Table 1 sensors-25-00586-t001:** Demographic data.

Variable	Median (IQR) or Number (%)
**Duration of Recording (min)**	4125.13 (1714.99–7250.14)
**Number of Patients**	109
**Age (years)**	43 (29–57)
**Sex (Male)**	89 (81.65%)
**GCS**	7 (4–8)
**GCS Motor**	4 (2–5)
**Pupils**	
**Bilateral Reactive**	65 (59.63%)
**Unilateral Reactive**	25 (22.93%)
**Bilateral Unreactive**	19 (17.43%)
**Marshall CT Score**	
**V**	55 (50.46%)
**IV**	20 (18.34%)
**III**	31 (28.44%)
**II**	3 (2.75%)

CT, computerized tomography; GCS, Glasgow Coma Score; IQR, interquartile range.

**Table 2 sensors-25-00586-t002:** Global population median optimal models for each resolution.

Temporal Resolution	ICP	AMP	RAP
**Minute-by-minute**	5, 1, 1	3, 1, 5	3, 1, 3
**10-min-by-10-min**	2, 1, 2	2, 1, 3	1, 1, 1
**30-min-by-30-min**	2, 1, 2	2, 1, 2	1, 1, 1
**Hour-by-hour**	2, 1, 2	1, 1, 1	1, 1, 1

AMP, pulse amplitude of ICP; ICP, intracranial pressure; RAP, compensatory reserve index.

**Table 3 sensors-25-00586-t003:** Global population median residuals of the signals.

Parameter	Original	Optimal ARIMA Model
**ICP**	0.63008	0.17941
**AMP**	0.49329	0.13549
**RAP**	0.32493	0.12564

AMP, pulse amplitude of ICP; ARIMA, autoregressive integrated moving average; ICP, intracranial pressure; RAP, compensatory reserve index.

**Table 4 sensors-25-00586-t004:** A comparative analysis between clean and artifact segments’ optimal ARIMA models (*p*-values from Mann–Whitney U-test).

Parameter	Minute-by-Minute		10-min-by-10-min	
	*p*-Order	*q*-Order	*p*-Order	*q*-Order
**ICP**	0.60996	**0.00058**	0.49778	**close to 0**
**AMP**	**0.00501**	0.20620	0.23175	**0.00032**
**RAP**	**close to 0**	**0.01526**	0.88808	0.94798

All the significant *p*-values are marked in bold. AMP, pulse amplitude of ICP; ARIMA, autoregressive integrated moving average; ICP, intracranial pressure; RAP, compensatory reserve index.

**Table 5 sensors-25-00586-t005:** Significant and insignificant counts after a Mann–Whitney U test between clean and artifact residuals.

Parameter	Minute-by-Minute		10-min-by-10-min	
	Significant	Insignificant	Significant	Insignificant
**ICP**	63	45	17	79
**AMP**	64	45	22	74
**RAP**	65	44	8	96

AMP, pulse amplitude of ICP; ICP, intracranial pressure; RAP, compensatory reserve index.

**Table 6 sensors-25-00586-t006:** Medians and means of the variance of the residuals.

Parameter	Minute-by-Minute				10-min-by-10-min			
	Median		Mean		Median		Mean	
	Clean	Artifact	Clean	Artifact	Clean	Artifact	Clean	Artifact
**ICP**	1.41843	24.83865	2.00069	266.45723	4.56886	35.47967	10.2683	1837.70208
**AMP**	0.04842	0.61611	0.08891	2.29946	0.11453	0.09194	0.27448	3.60461
**RAP**	0.10523	0.20676	0.10882	0.21222	0.08715	0.10794	0.09471	0.11736

AMP, pulse amplitude of ICP; ICP, intracranial pressure; RAP, compensatory reserve index.

**Table 7 sensors-25-00586-t007:** Medians and means of the maximum cross-correlations of residuals.

Parameter	Minute-by-Minute				10-min-by-10-min			
	Median		Mean		Median		Mean	
	Clean and Clean	Clean and Artifact	Clean and Clean	Clean and Artifact	Clean and Clean	Clean and Artifact	Clean and Clean	Clean and Artifact
**RAP-ICP**	14.48875	23.01389	77.319716	46.14512	26.92	32.46088	36.9293	121.002
**RAP–AMP**	137.63467	28.61530	415.91384	51.74234	4.4332	1.55919	5.83130	3.97946

Clean and clean refers to the cross-correlation between clean RAP and clean ICP (RAP–ICP) or clean RAP and clean AMP (RAP–AMP), whereas clean and artifact refers to clean RAP and artifact ICP (RAP–ICP) or clean RAP and artifact AMP (RAP–AMP).

## Data Availability

Data are contained within the article.

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
