# Peer review of "Characterization of RAP Signal Patterns, Temporal Relationships, and Artifact Profiles Derived from Intracranial Pressure Sensors in Acute Traumatic Neural Injury"

_sensors, 2025, doi:10.3390/s25020586_

Round 1
Reviewer 1 Report
Comments and Suggestions for Authors
In this study, the authors applied RAP (the moving correlation between intracranial pressure (ICP) and the pulse amplitude of ICP (AMP)) to analyze the cerebral compliance of patients with moderate to severe traumatic brain injury (TBI) and validated the performance of the proposed model. The methodologies and results are generally well structured and appropriately intepreted. Limitations are also adequately addressed to initiate further improvement. The work contribute to advancing bedside TBI monitoring in a more statistically robust way and can be improved by addressing the following questions:
1. In the manuscript, it would improve the significance of the proposed method and results if other similar work or model is compared. Why choose median optimal ARIMA model? Is there additional model examined?
2. For the paragraph starting at line 441, it seems that the analysis performed at different temporal resolutions only included minute-by-minute and hour-by-hour cases. It is suggested to justify the choice or extend the comparison in the context for clarification. Similarly, in section 3.4. for the artifact assessment, it would better to clarify the different choices of resolutions.
3. Please check the consistency of the heading numbers, e.g., "3.2.2. Determination of Optimal ARIMA Models" and "3.2.3. AMP, Pulse Amplitude of ICP; ICP, Intracranial Pressure; RAP, Compensatory Reserve Index".
Author Response
Reviewer 1
- In the manuscript, it would improve the significance of the proposed method and results if other similar work or model is compared. Why choose median optimal ARIMA model? Is there additional model examined?
Thank you for highlighting this issue.
ARIMA was chosen since it is ideal for capturing the autoregressive, differencing, and moving average components of physiological signals, allowing for direct comparisons between clean and artifact data to identify deviations caused by artifacts (assessing all the linear relationship aspects of a signal). Unlike alternative methods, such as frequency-domain or machine learning models, ARIMA provides interpretability in terms of temporal dependencies (p, d, q), making it particularly suited for understanding signal dynamics, comparing temporal structure among signals and identifying artifacts.
Additionally, the median values were selected for the optimal models, because we had comparatively a small dataset. This approach ensured a more representative summary across the examples while reducing the impact of outliers. Moreover this is similar methodology to the assessment of more basic physiologies like ABP and ICP (PMID: 32418148 and https://doi.org/10.1016/j.bspc.2024.106403)
The following lines were added addressing this issue in the second paragraph of section 2.6.1:
ARIMA was chosen since it provides interpretability in terms of temporal dependencies (p, d, q), making it particularly suited for understanding signal dynamics, comparing temporal structure among signals and identifying artifacts. As well as it is in similar methodology to the more basic aspects of cerebral blood flow physiologies.
And in section 2.6.5
The choice of median value for optimal models across the whole population ensured a more representative summary across the examples while reducing the impact of outliers.
- For the paragraph starting at line 441, it seems that the analysis performed at different temporal resolutions only included minute-by-minute and hour-by-hour cases. It is suggested to justify the choice or extend the comparison in the context for clarification. Similarly, in section 3.4. for the artifact assessment, it would better to clarify the different choices of resolutions.
Thank you for your valuable feedback regarding this issue. For the paragraph starting at line 441, the following changes were made to the figure 2 of the manuscript and figure 4 of Appendix E:
Figure 2: ACF and PACF plots at different resolutions – patient example
(a) At 10min-by-10min resolution with ARIMA (1, 1, 1) (b) At 30min-by-30min resolution with ARIMA (1, 1, 1)
(c) At hour-by-hour resolution with ARIMA (1, 1, 1)
The figure documents the ACF and PACF of the residuals of RAP-mapped ARIMA structure in the (a) 10 minute-by-10 minute, (b) 30 minute-by-30 minute, and (c) hour-by-hour relationships.
Figure 2 ACF and PACF plots for ICP at different resolutions for individual
a) At 10min-by-10min resolution with ARIMA (2, 1, 2) (b) At 30min-by-30min resolution with ARIMA (2, 1, 2)
(c) At hour-by-hour resolution with ARIMA (2, 1, 2)
The figure documents the ACF and PACF of the residuals of ICP-mapped ARIMA structure in the (a) 10 minute-by-10 minute, (b) 30 minute-by-30 minute, and (c) hour-by-hour relationships.
Figure 4 ACF and PACF plots for AMP at different resolutions for individual
a) At 10min-by-10min resolution with ARIMA (2, 1, 3) (b) At 30min-by-30min resolution with ARIMA (2, 1, 2) (c) At hour-by-hour resolution with ARIMA (1, 1, 1)
The figure documents the ACF and PACF of the residuals of AMP-mapped ARIMA structure in the (a) 10 minute-by-10 minute, (b) 30 minute-by-30 minute, and (c) hour-by-hour relationships.
Several edits were made to the paragraph (starting at line 441), which now reads as follows:
A similar analysis was performed at different temporal resolutions for the same patients. Figure 2 shows a patient example of the ACF and PACF plots of the residuals for RAP at the remaining temporal resolutions. Even though all the lags were within the 95% confidence interval at all the resolutions (which was also justified by the result in Table 2 of Appendix E), the ACF and PACF plots exhibited comparatively significant spikes with higher magnitude than those at the minute-by-minute temporal resolution (depicted in Figure 1(b)). This indicates that while the optimal ARIMA model captures the time-series structure in both cases, it is more successful at the minute-by-minute resolution than at the other lower temporal resolutions. Figure 2 and Figure 4 of Appendix E present these comparative figures for ICP and AMP at the 10-minute, 30-minute and hour intervals, which also displayed similar characteristics in the ACF and PACF plots of the residuals.
For section 3.4, the analysis on identifying artifacts only included minute-by-minute and 10-minute intervals. This is because the 30-minute and hour-by-hour intervals were excluded because, at such low resolutions, the quantity of artifact data would be insufficient to yield significant results in this section. The following lines were added in the last sentence of the first paragraph of section 2.7.2.
The remaining resolutions were excluded from this analysis because, at such low resolutions, the quantity of artifact data would be insufficient to yield significant results in this section.
- Please check the consistency of the heading numbers, e.g., "3.2.2. Determination of Optimal ARIMA Models" and "3.2.3. AMP, Pulse Amplitude of ICP; ICP, Intracranial Pressure; RAP, Compensatory Reserve Index".
Thank you for pointing out this. The headings were corrected to:
3.3.2. Determination of Optimal ARIMA Models and 3.3.3. Evaluation of Optimal ARIMA Models.
Reviewer 2 Report
Comments and Suggestions for Authors
This is a very interesting stuy by a well known group regarding predictive modelling of ICP elevation in patients with TBI using different physiomarkers of craniospinal dynamics. I do not have any comment on the methodology. My only concern is if the authors evaluated how their model works upon different therapeutic interventions, such as decompressive craniectomy, pCO2 changes or mannitol infusion. I think that such 'stress tests' of ICP manipulation must be tested for increased validity and accuracy of future prognostic modelling.
Author Response
Reviewer 2
This is a very interesting stuy by a well known group regarding predictive modelling of ICP elevation in patients with TBI using different physiomarkers of craniospinal dynamics. I do not have any comment on the methodology. My only concern is if the authors evaluated how their model works upon different therapeutic interventions, such as decompressive craniectomy, pCO2 changes or mannitol infusion. I think that such stress tests; of ICP manipulation must be tested for increased validity and accuracy of future prognostic modelling.
Thank you for the thoughtful comments surrounding the impact of ICP mediation treatments and their subsequent impact on RAP. We have done some preliminary work assessing the extreme levels of ICP and AMP (see Appendix A), which will likely be linked to the aforementioned ICP treatments. However, we do agree that future work exploring the impact of various critical care treatment of waveforms is required, thus we have added the following to the limitations section.
To address this, the following lines were added in section 5 (limitation):
Fourthly, a detailed treatment based sub-group assessment is required. This analysis did not include the effect of different therapeutic interventions, such as decompressive craniectomy, pCO2 changes or mannitol infusion. Though ICP treatments have an immediate impact on ICP (the minutes after treatment), their long-term impact on ICP modelling and other derived ICP measures (like PRx) is quite limited. Therefore, when modeling and assessing RAP physiological factors over larger periods of time and over whole populations, ICP treatment factors can likely be largely ignored. However, when robust minute-by-minute RAP is being modeled (looking at individual moments of patient state); these factors should be considered.
Round 2
Reviewer 1 Report
Comments and Suggestions for Authors
The questions have been properly addressed.